# Fault Structural Analysis Applied to Proton Exchange Membrane Fuel Cell Water Management Issues

**Etienne Dijoux** [1,2,*], **Nadia Yousfi Steiner** [2], **Michel Benne** [1], **Marie-Cécile Péra** [2] and **Brigitte Grondin-Perez** [1]

1   ENERGY Lab—LE2P, University La Reunion, 97415 Saint-Denis, France;
    michel.benne@univ-reunion.fr (M.B.); Brigitte.grondin@univ-reunion.fr (B.G.-P.)
2   FEMTO-ST Institute, FCLAB, University Bourgogne Franche-Comté, 90000 Belfort, France;
    nadia.steiner@univ-fcomte.fr (N.Y.S.); marie-cecile.pera@univ-fcomte.fr (M.-C.P.)
*   Correspondence: etienne.dijoux@univ-reunion.fr

**Abstract:** Proton exchange membrane fuel cells are relevant systems for power generation. However, they suffer from a lack of reliability, mainly due to their structural complexity. Indeed, their operation involves electrochemical, thermal, and electrical phenomena that imply a strong coupling, making it harder to maintain nominal operation. This complexity causes several issues for the design of appropriate control, diagnosis, or fault-tolerant control strategies. It is therefore mandatory to understand the fuel cell structure for a relevant design of these kinds of strategies. This paper proposes a fuel cell fault structural analysis approach that leads to the proposition of a structural graph. This graph will then be used to highlight the interactions between the control variables and the functionalities of a fuel cell, and therefore to emphasize how changing a parameter to mitigate a fault can influence the fuel cell state and eventually cause another fault. The final aim of this work is to allow an easier implementation of an efficient and fault-tolerant control strategy on the basis of the proposed graphical representation.

**Keywords:** fuel cell fault structural analysis; diagnosis; fault tolerant control; fuel cell cathode water management





## 1. Introduction

Proton exchange membrane fuel cells (PEMFCs) are efficient and clean energy supply systems. However, they are subject to the occurrence of various faults, which decreases their reliability. Faults are inordinate phenomena that degrade a system's performance more or less rapidly and substantially [1]. Their occurrence can be attributed to several factors (exogenous and endogenous). Both exogenous factors, such as gas purity or demanding load profile, and endogenous factors, such as poor internal design or natural aging, can lead to fault occurrence and, therefore, to fuel cell damage. The operating conditions need to be adjusted to mitigate the faults. Moreover, PEMFCs are nonlinear, multivariate, and strongly coupled systems, which complicates their ability to be maintained under normal operation. Therefore, it is necessary to highlight the coupling of the PEMFCs' parameters to facilitate the understanding of the fault occurrence process. Indeed, an exhaustive analysis of the variables' effects and interactions inside the system is a major issue to be considered to set an efficient fault-tolerant control (FTC) strategy.

The literature provides different approaches to modeling and analyzing a system in order to understand the influences of its variables and their interactions. Noyes [1] highlighted two types of methodologies that allow this analysis: statistical and functional. On the one hand, a statistical analysis consists in the observation of events [2,3] with the aim of making assumptions to predict events in similar situations. For instance, Bayesian statistical analysis [2] aims to process small datasets. Indeed, this approach allows one to obtain relevant information by limiting costly observations. The obtained information is then iteratively refined according to a Bayesian law. On the other hand, functional analysis,

focuses on all functions ensured by the system and their influence on the occurrence of a fault. These approaches are usually based on a graphical formalism that leads to the specification of good operating conditions.

Several methods for functional analysis can be found in the literature, such as FAST (Function Analysis System Technique), SADT (Structured Analysis and Design Technique), or FTA (Fault Tree Analysis).

FAST is a graphical representation of a system's functions that answers the following questions: why does a function have to be ensured? How is this function ensured? When must the function be ensured? Other functional analysis methods also exist in the literature [4–6], and they consist in analyzing a system with measurements to observe its structural and functional modifications.

The SADT [7,8] method is a graphical tool associated with a top-down analysis method. This method is used to decompose a function with a functionally oriented methodology. It consists of modeling the process by breaking it down into subsets. A data-based or function-based diagram is then created to model the process of each subset.

FTA is also a functionally based analysis that allows one to perform a failure mode analysis. It is a top-down approach: a top event is considered, and all combinations of sub-events that lead to it are determined [9].

The top event is reached with a combination of several sub-events. Sub-events have their origin in the combination of basic events. Fault trees, therefore, bring information about the variables (basic events) that are involved in a specific fault occurrence.

To summarize, on the one hand, FAST uses a diagram to organize the ways of thinking, acting, or talking. It enables the development of technical solutions according to a functional logic. However, this method does not consider the system complexity or the coupling phenomena of internal system variables. On the other hand, the SADT method takes into consideration the complexity and allows the analytical decomposition of the system according to a hierarchical structure. However, this approach does not allow links between transitions of operating conditions, such as from normal condition to a faulty one. On the contrary, FTA is relevant in understanding how a fault can occur with a combination of basic events. Indeed, FTA allows the link between each transition of the operating conditions. With logic gates and sub-events, it describes all paths that lead to the appearance of the fault. FTA highlights the information about fault occurrence and, therefore, the relevant variables for fault mitigation. Fuel cells are strongly coupled systems, and several variables have mutual interactions that are not highlighted by FTA. For this reason, a new analysis methodology, which is called Fault Structural Analysis (FSA), is applied to these systems. Indeed, the FSA allows describing the system's structure and highlighting all variables that influence the fuel cell operating conditions. This approach is also relevant for designing fault tolerant control strategies because it is helpful for fault mitigation and system monitoring process. For instance, as presented in [10], authors present their strategy for fuel cell fault mitigation. Their work consists of gathering information about the PEMFC state of health through the remaining useful lifetime. The approach is based on the analysis of the system nominal and faulty conditions which are provided by a key variable behavior. This strategy is thus highly dependent on the relevant choice of the key variable that should be subject to a study of its field of action in the fuel cell for more efficiency. In [11], Yang et al. try to improve the PEMFC reliability with the implementation of a robust fault observer for air management system fault diagnosis. Once again, the choice of the estimated variable is a key factor for their diagnosis tool. Indeed, the implementation of their strategy depends on sensitivity of the diagnosis variables to the fuel cell functionalities which are subject to faulty conditions. In [12], authors proposed a fuel cell health management system. They used the electrochemical impedance spectroscopy (EIS) in a fault tolerant control strategy in order to diagnose the water management faults. The drawback of EIS diagnosis tools lies in their low computational time, their offline operating mode and the cost of the used equipment. To avoid this problem, a solution for the implementation of a diagnostic tool can be based, for each relevant variable, on identifying the one most influenced by each

faulty condition. Another study proposed by Rubio et al. [13], consist of the implementation of a fuzzy model to determine the water dehydration in a PEMFC. The real-time aspect of the strategy involves the use of fast response time of the control variable. The current, the flow rate and the voltage are thus used in the strategy for the fuel cell hydration characterization. This study only considers fast response time variables for the diagnosis tool, but the studied phenomena have low, medium and high frequency behavior. In the case of the introduction of variables which are influenced on the overall spectrum, authors would improve their strategy efficiency.

The FSA design leads to a graphical representation which is achieved in three steps: (i) identification of the system's functionalities and definition of each control variable, (ii) finding the constraints (restriction of the system functionalities) that are influenced by system's variables and (iii) designing the structural graph.

Water management faults is a recurrent faulty condition for PEMFC systems. Indeed, they can lead to severe performance losses and in some cases to irreversible degradation. Therefore, the FSA approach that leads to a structural graph considers two faulty conditions related to the water management: flooding and membrane drying out. The introduction of these two faults in the structural graph will underline the fuel cell functionalities impacted by their occurrence, and also highlight the available control variables which can be used for their mitigation in the case of an FTC strategy. FSA leads thus to the following contributions:

- Describe the fuel cell structure only with a graph,
- Highlight all variables which influence the fuel cell functionalities and therefore its operating conditions,
- Underline the links between the fuel cell functionalities and faults,
- Highlight the relevant control variables which can be used for fault mitigation

This paper is organized as follows: The first section is dedicated to the PEMFC water management issues. The second section proposes a structural analysis which allows highlighting all couplings between the system variables. Then, in the third section the structural analysis approach is defined and applied on the PEMFC system. The two last sections discuss and conclude about the structural analysis approach applied to a PEMFC system.

## 2. Water Management Faults

PEMFC systems may be subjected to different faulty operating modes. In [14], authors define fault as a decrease in system performance caused by improper major or minor fuel cell operation. These kinds of operations could lead to a permanent loss of the fuel cell performance due to the occurrence of faults or fuel cell ageing. This paper only focuses on faults which lead to fuel cells performance losses. These faults can be classified according to some criteria like: effects, response time, recovery property (the loss of performance can be totally recovered or not) or location.

Fuel cell faults can be detected and isolated, with the use of appropriate fault diagnosis tools. The literature relates several diagnosis techniques and many of them are used for PEMFCs water management issues. For instance, Lu et al. [15] proposed a fault diagnosis based on a fast electrochemical impedance spectroscopy (EIS) measurement. The developed tool allows an on-line flooding and drying out diagnosis but the authors underline that it cannot rules on the fuel cell state of health (SoH) in the case of multiple fault occurrence. Regarding their experiment, the multiple fault occurrence happens when the recovering time between each faulty condition is not enough. Therefore, in order to have a complete fault deletion and to improve the diagnosis tool performance, it is therefore relevant to know what the involved variables during a fault mitigation process are. Another example of a diagnosis tool consists of a model-based observer for fuel cell internal states estimation [16]. Authors aim to resolve the unmeasured internal variables issue thanks to a virtual sensor based on observer. This work highlights the need to understand the fuel cells operation mechanism through the internal state estimation and by the coupled variables involved in the change of the fuel cell SoH. In the follow-up, Alves-Lima et al. [17]

have worked on a quantitative video-rate hydration imaging of Nafion. Their results have shown that membrane water content is correlated to several factors: the membrane thickness, the fuel cell temperature (the room temperature in their study) and the water desorption process are major factors that influence the membrane hydration. Once again, the knowledge of the fuel cell internal variables are major issues to maintain the membrane under nominal hydration. In [18] authors use an EIS measurement to characterize the impact of the membrane water management on the performance of PEMFC commercial stacks. Their goal is to understand what the effects of the inlet gas water content on the fuel cell operations via the analysis of the Nyquist plot for fuel cell stack and single cells are.

These papers underline the need of knowledge about the internal fuel cell variables and how they have mutual influences. Indeed, as shown in [17], a modification of only one internal variable value could lead to the system destabilization. For this reason, the FSA analysis is focused on all the variables involved in the fuel cell to highlight extensively and systematically the multilateral effects of water management faults inside the fuel cell.

Cell flooding and membrane drying out are two possible consequences of improper water management. Flooding is caused by an accumulation of liquid water either in the diffusion layer of the electrodes, the bipolar plate channels or the feeding lines that limits the access of the reactants to the catalyst sites, and then decreasing the electrochemical reaction rates. Membrane drying out is the result of an insufficient hydration of the PEMFC membrane, thus increasing its proton resistivity.

Li et al. [19] identified flooding as the most recurrent PEMFCs' fault and point out that the cathode—being the place of water production—is particularly affected. The antagonist phenomenon is the drying out. It can occur when the gases relative humidity is too low [20,21], the input gas flow rate is too high, the operating temperature is too high or when these improper operating parameters are combined.

To better understand these faults, a functional analysis of PEMFC water management has been investigated through the Fault Tree Analysis in [11], among the possible approaches presented previously. The FTA is reported in Figure 1.

The FTA approach provides information about some coupling phenomena between variables inside the system. For instance, the temperature (T) influences the gas relative humidity. However, it is also linked to other variables which can be used to settle a fault tolerant control law. It is therefore important to understand how the different control variables are coupled to reduce the risk of unexpected phenomena. The proposed FSA approach aims to bring out this information in an explicit and systematic way via the design of diagrams called structural graphs.

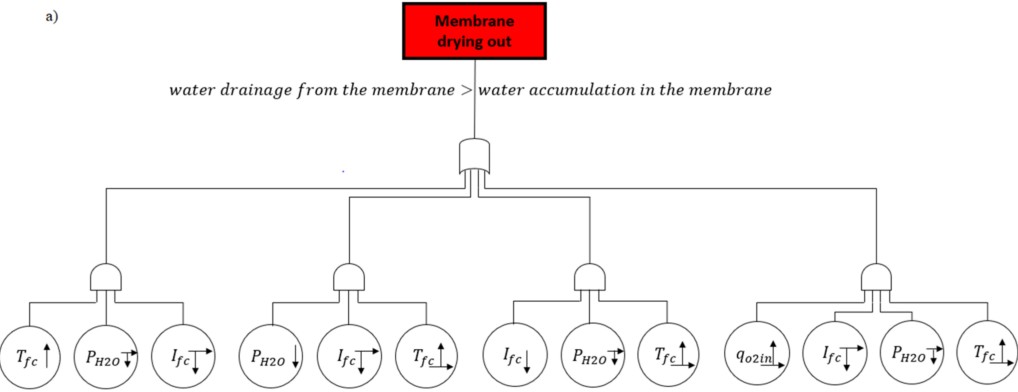

**Figure 1.** *Cont.*

b)

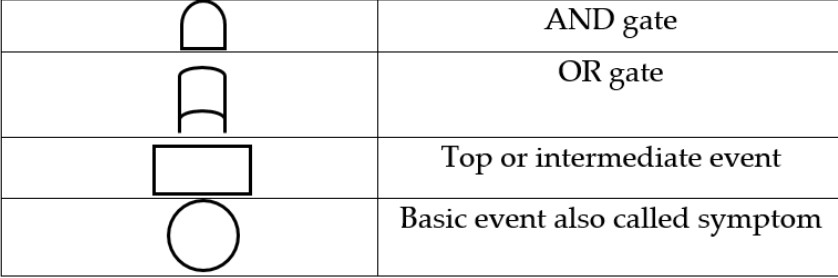

| | |
|---|---|
| ⌓ (AND gate symbol) | AND gate |
| ⌒ (OR gate symbol) | OR gate |
| ▭ (rectangle) | Top or intermediate event |
| ◯ (circle) | Basic event also called symptom |

**Figure 1.** FTA applied on membrane drying out (**a**) and on a flooding (**b**) reproduced with the permission from Yousfi Steiner et al., 2021 [9].

### 3. Fault Structural Analysis: Definition and Objectives

FSA is a low-level representation of a system behavior that allows highlighting the operating conditions that potentially lead to a fault occurrence and the variables that could be used to mitigate this fault. The representation is based on connections between the system variables through a bipartite graph. All system features are described by a set of constraints, viewed as restrictions on the system functionalities, and the violation of one of them that indicates a fault occurrence.

*3.1. Dynamical Systems*

Structural analysis considers only the structural information, highlighting the variable involved in the studied phenomenon. A representation is given in Figure 2.

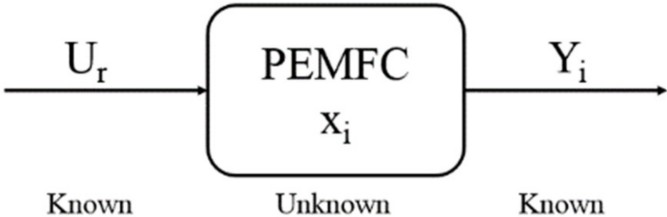

**Figure 2.** Known and unknown variables representation.

Inputs $U_r$ are defined as: $U_r$ with $r \in \{1,2, \dots ,n\}$
Outputs $Y_i$ are defined as: $Y_i$ with $i \in \{1,2, \dots ,k\}$
Unknown states, which are not directly measured but could be estimated from the known ones, they are defined by: $x_j$ with $j \in \{1,2, \dots ,l\}$.

### 3.2. Bipartite Graph

Beauguitte [22] defined the bipartite graph as a structure that displays relationships between two separate sets of vertices, describing system characteristics, variables, and constraints. For this reason, in the next section the bipartite graph will be called structural graph. The author underlines that vertices can be separated according to their contribution to an event such as the occurrence of a particular fault.

The links between vertices are usually not oriented and represent the system's structure which is noted as: $G = (C \cup Z, \Gamma)$, where G is the structure of the bipartite graph and U the union operator. Z is the set of characteristic variables: $Z \{z1, z2, \dots , zn\}$. C is the set of constraints: $C = \{c1, c2, \dots , cm\}$. Arcs that connect each vertex (constraints/variables) are noted:

$$\Gamma = \{(c\_i, z\_j) \dashv \mid z\_j \text{ exist in } c\_i, z\_j \in Z, i \in [1,m], j \in [1,n]\}. \tag{1}$$

An illustration of a bipartite graph is given in Figure 3.

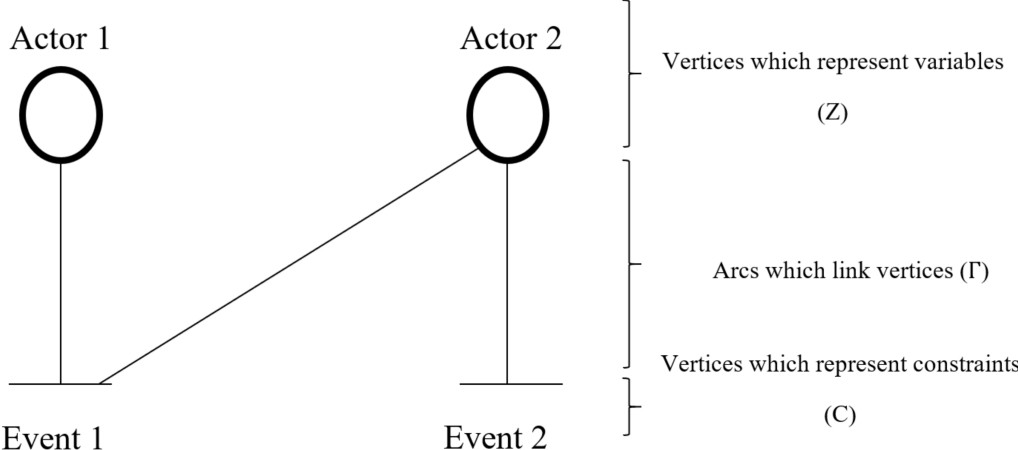

**Figure 3.** Illustration of a bipartite graph.

It is possible to create an incidence matrix M that represents the bipartite graph and consists of a Boolean matrix where the rows are the constraints and the columns the characteristic variables:

$$M = \{m_{i,j} \mid m_{i,j} = 1 \text{ if } (c_i, v_j) \in \Gamma, 0 \text{ else}\} \tag{2}$$

### 3.3. Differentiation

The studied phenomena (faults) are time dependent. Therefore, a dynamic model is mandatory in order to take into account time dependent variables. Three options can be considered:

Option 1: Considering x (variable of the studied system) and $\dot{x}$ as the same variable and treating the dynamical equations in the same way as the static ones.

Option 2: Considering x and x = dx/dt as structurally distinct and using the model with the explicit differential equation: dx/dt ([23]).

Option 3: Considering x and x = dx/dt as structurally distinct and proceeding to the structural differentiation of the initial model ([24]).

The choice to use the explicit differential equation $\dot{x}$ is preferred in our study to take into consideration the system dynamic behavior with the present and past values (option 2).

*3.4. Representation of Faults*

As said before, faults are abnormal phenomena which decrease the system performance and can lead to its degradation. It is thus mandatory to detect their occurrence to proceed to a fault mitigation strategy. In case a diagnosis tool is used, several levels of a priori knowledge must be considered.

- A low level of knowledge, that only allows specifying the system functionalities (i.e., the function which describes the operations) which are influenced by the faults' occurrences. The description of the cause-and-effect relationship is not mandatory.
- A medium level of knowledge for which an analytical description of the cause-and-effect relationships between system functionalities and faults is available. In this case, variables that describe the faults must be integrated into the model.
- A high level of knowledge for which a fault model is specified.

For this work, a low level of knowledge is considered. Indeed, accurate fault model is not mandatory. Only the functionalities that are influenced by their occurrence are needed whereas the cause-and-effect relationship is not.

The FSA approach and objectives are now explained. The structural graph design process is therefore described by the following steps: (i) find a model of the system which describes its functionalities, (ii) identify the system constraints in order to get all the variables which have an influence on the system operation, (iii) create an incidence matrix for the design of the structural graph. The next section consists in choosing a PEMFC model to guide the structural graph design.

## 4. PEMFC Functionalities and Control Variables

*4.1. PEMFC System*

A PEM fuel cell is composed of an anode, supplied with hydrogen, and a cathode, supplied with oxygen (pure or from the air). At the anode side, one molecule of hydrogen is oxidized thanks to a catalyst made of platinum (Pt). It allows the formation of two protons and two electrons:

$$H_2 \rightarrow 2H^+ + 2e^- \tag{3}$$

At the cathode side, oxygen is reduced by the protons thanks to a catalyst made of platinum (Pt). Its reduction allows the formation of water through the electrochemical reaction:

$$1/2\,O_2 + 2H^+ + 2e^- \rightarrow H_2O \tag{4}$$

The electrons move from the anode to the cathode through an external electric circuit. These two electrodes are separated by a proton exchange membrane that is impermeable to reactant gases. The supply of reactants to the PEMFC and the exhausts are carried in and out via channels in bipolar or mono-polar plates as represented in [25].

For the next sections, the studied system is a PEMFC stack. All cells are supposed to have the same mean behavior, and the reactant supply system is represented on the scheme on Figure 4:

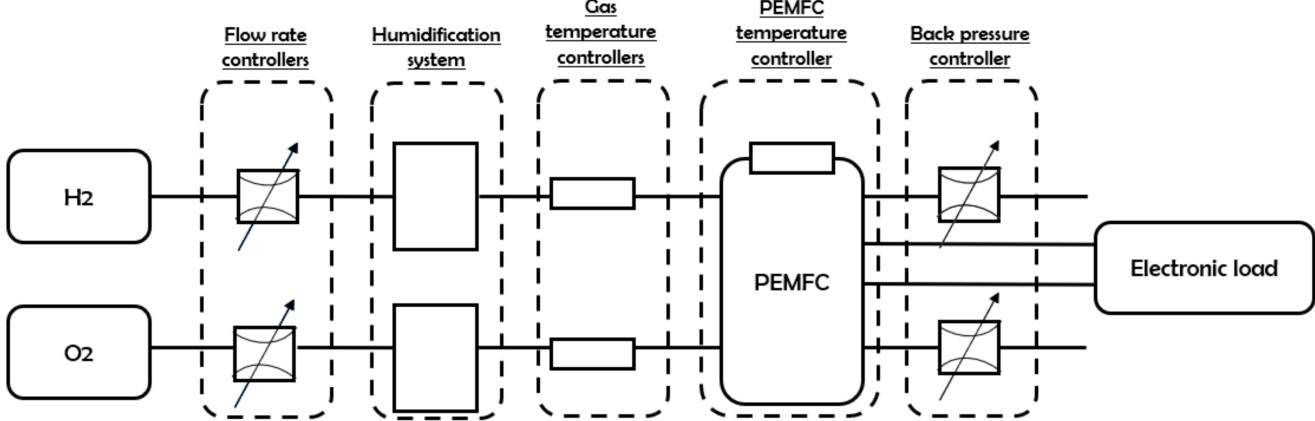

**Figure 4.** Example of humidified system synoptic of a PEMFC.

The Figure 4 represents the main elements of the PEMFC system that will be involved in fault generations and then monitored for their mitigation. Indeed, the system is supplied with hydrogen and oxygen and the gas flows are controlled with two controllers which manage to supply the PEMFC with the good ratio of reactants. The reactants are humidified with the humidification system. The PEMFC input gas relative humidity is adjusted with a heated line. The PEMFC pressure is controlled with two back pressure controllers. To simplify the representation, in the following, the considered oxidant is pure oxygen, but the approach would be the same with air.

*4.2. PEMFC Modeling with an Energetic Macroscopic Representation (EMR)*

To perform an FSA for PEMFCs water management faults, a zero-D fuel cell model is considered. The energetic macroscopic representation (EMR) proposed in [26] is used to identify the system functionalities. This macroscopic model is chosen because it highlights the system's control variables. Indeed, the structural analysis presented in this paper aims to bring the relevant control variables which could mitigate a fault occurrence. Other multidimensional models would lead the same analysis, but they involve more variables which are not measurable or controllable. Therefore, this kind of model are useless for the aim of fault mitigation.

The EMR representation is a quasi-static model involving differential equations that make possible the study of time-dependent systems. It also specifies the equations that describe the system's functionalities that could be submitted to a faulty condition. The principle of the EMR consists in considering the different power conversions and the cause-consequence relationship. Then, each element is linked to the others through a couple of action/reaction variables which product is a power. The instantaneous conversion and the accumulation of power are distinguished.

The PEMFC is modelled as several sub systems linked together in order to create a complete fuel cell model. This simple model and few additional equations are relevant to perform the ASD because it allows taking into account the variables of interest.

This representation is composed of three parts.

First, the fluidic part is dedicated to the gas channels located between the gas tank and the reaction sites. An electric analogy is used with pressure is assumed to be a potential and volume flow rates are assumed to be currents. Using electrical analogy:

- the system is also composed of a distribution resistance RdO21, an exhaust resistance RdO22 and a hydraulic capacity Ch (for gas accumulation in the circuit),
- and the current generator represents the consumption of the reactant or the generation of the products.

The Figure 5 shows the analogy between the pipe and the RC electrical circuit [27].

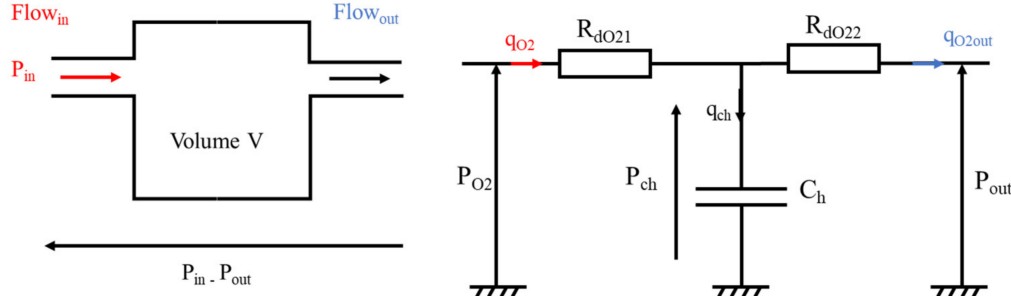

**Figure 5.** Analogy between pipe of the hydraulic circuit and an electric RC circuit.

The oxygen circulation qO2 is enabled by fuel cell input/output pressure difference. The consumed (resp. produced) gases' flows qcO2 are calculated from the Faraday law and the number of electrons exchanged nO2 counted negatively (resp. positively). Partial pressures at each electrode are imposed by the gas accumulation represented by qch. Ifc is the electric current of the fuel cell, N is the number of cells.

$$PO2 = PscO2 + RdO21\ qO2 \tag{5}$$

$$qO2out = (PscO2 - PsO2)/RdO22 \tag{6}$$

$$(dPscO2)/dt = 1/Ch\ (qO2 - qcO2 - qO2out) \tag{7}$$

$$qcO2 = \pm N * Ifc/(nO2 * F) * (R * Tfc)/PO2 \tag{8}$$

$$qO2out = qO2 + qcO2 \tag{9}$$

In Equation (7) the derivative term is assumed to be structurally different from PscO2 (Cf. Section c).

The second part of the EMR is the electrochemical part. The Nernst potential En is based on the computation of a thermodynamic potential E0 and the influence of the partial pressures and the temperature:

$$En = E0 + \Delta E \tag{10}$$

with,

$$E0 = \alpha + \beta * Tfc + \gamma * Tfc\char`\^2 + \delta * Tfc\char`\^3 + \nu * Tfc * \ln(Tfc) \tag{11}$$

and,

$$\Delta E = Acd * \ln (PscH2/P0) + Bcd * \ln(PscO2/P0) \tag{12}$$

where $\alpha$, $\beta$, $\gamma$, $\nu$, Acd(Tfc) and Bcd(Tfc) are model adjustment variables. Tfc is the fuel cell temperature and the chosen model does consider a cooling system. E0 is the thermodynamic potential of the PEMFC [26]. The voltage drop $\Delta V$ calculation (activation, concentration and ohmic losses) is then carried out:

$$\Delta V = ⟦\Delta V⟧\_act + ⟦\Delta V⟧\_conc + ⟦\Delta V⟧\_ohm \tag{13}$$

The PEMFC voltage is thus:

$$V\_M = E\_n - \Delta V \tag{14}$$

Each loss is expressed as follows, using In the cross over current, I0 the exchange current, and Il the limit current:

$$\Delta Vact = A * Tfc * \ln((Ifc + In)/I0) \tag{15}$$

$$\Delta Vconc = B * Tfc * \ln(1 - Ifc/Il) \tag{16}$$

$$\Delta Vohm = Rm * Ifc \tag{17}$$

The third part is the electric impedance of the cell, represented by the block "Charge double layer" but as the fast-electric dynamic is not considered in the analysis, this block is not considered. Considering N the number of stack cells, the stack voltage is expressed as follows:

$$\text{Vfc} = N * (VM + Vc) \tag{18}$$

Thanks to this fuel cell model, it is possible to define the control variables for the system functionalities. All variables that influence the PEMFC functionalities are supposed to be known. The next step consists of defining the constraints that can be influenced by a fault occurrence.

## 5. Constraints of PEMFC

### 5.1. Structural Analysis of a PEMFC

The structural analysis design is focused on the cathode fluidic part because it is the location of the water production. The cathode area therefore represents the major issue regarding fuel cells water management. It is located between the gas tank and the reaction sites. This part is modeled with the electric analogy represented on Figure 5. The oxygen circuit is composed of a fluidic resistance $Rd_{O21}$, an exhaust resistance $Rd_{O22}$, a hydraulic capacity $Ch_{O2}$ and the consumed oxygen flow generator. The water circuit is composed of the same type of elements, with the produced oxygen flow generator. The input gas flow $q_{O2}$ is imposed by a flow controller, the values of the input water flow $q_{H2Oin}$ is imposed by the humidification system depending on the controlled humidity rate and the temperature of the fuel cell $T_{fc}$. The pressure at the exhaust is the atmospheric pressure.

The gas flow inside the PEMFC supply channels is considered as the first event for the normal fuel cell operation. This first event, called constraint for the structural analysis, is the first constraint noted C1. However, this gas circulation can be influenced by the system variables given in the following equations:

$$P_{O2in} - P_{O2out} = R_{dO21} * q_{O2in} + Rd_{O22} * q_{O2in} - q_{O2ca} \tag{19}$$

where $P_{O2in}$ and $P_{O2out}$ are the pressure of the input and output gas respectively. Regarding the water circuit the equation becomes:

$$P_{H2Oin} - P_{H2Oout} = R_{dH2O1} * q_{H2Oin} + R_{dH2O2} * q_{H2Oin} + q_{H2Oca}$$

The cathode pressure drop is expressed as follows:

$$\Delta P = P_{tot\_in} - P_{tot\_out}$$

Using the humidified input gas flow, which is actually the total input flow ($q_{O2hum\_in}$), the pressure drop expression becomes:

$$\Delta P = R_{dO2hum_1} q_{O2hum_{in}} + R_{dO2hum_2} \left( q_{O2hum_{in}} - q_{O2_{ca}} + q_{H2O_{ca}} \right)$$

Thus:

$$C1: \ q_{O2hum_{in}} = \frac{1}{R_{dO2hum_1} + R_{dO2hum_2}} \left( \Delta P + R_{dO2hum_2} q_{O2_{ca}} - R_{dO2hum_2} q_{H2O_{ca}} \right) \tag{20}$$

The humidified gas inside the channels crosses the gas diffusion layer (GdL) for the purpose of air diffusion to the catalytic site. However, the GdL gas concentration has to be kept at a high value for a good and safe catalytic feeding. The gas amount accumulated in the GdL is thus the third constraint C2. The variables which have an influence on this gas amount inside the GdL appear in the expression of the hydraulic capacity of the GdL Ch:

$$C_h = \frac{1}{P_{O2_{ca}} + P_{H2O_{ca}}} \left( q_{O2} + q_{H2O_{in}} - q_{cO2} + q_{H2O_{ca}} - q_{O2_{out}} - q_{H2O_{out}} \right) \tag{21}$$

where:

$$q_{O2hum_{out}} = \frac{P_{O2_{ca}} + P_{H2Oca} - P_{O2_{out}} - P_{H2O_{out}}}{R_{dO2hum_2}} \tag{22}$$

hence:

$$C2: C_h = \frac{1}{P_{O2_{ca}} + P_{H2O_{ca}}} \left( q_{O2_{in}} + q_{H2O_{in}} - q_{O2_{ca}} + q_{H2O_{ca}} - \frac{P_{O2_{ca}} + P_{H2Oca} - P_{O2_{out}} - P_{H2O_{out}}}{R_{dO2hum_2}} - q_{H2O_{out}} \right) \tag{23}$$

It should be noted that the constraint C2 has also the fuel cell temperature as input variable. Indeed, the flowrates are expressed as volume so they depend on Tfc.

At the catalytic sites, and for high current densities, the kinetic of reactions increases and the gas consumption intensifies. During this operating condition, the quantity of O2 species decreases and the steam production increases. Therefore, the fuel cell voltage decreases. The steam partial pressure at the catalytic site is thus the fifth constraint C3 because it has to be under the saturation pressure value to avoid water condensation. The variables which influence this constraint are expressed by the following inequality relationship which represent the steam partial pressure at the catalyst (PH2Oca) and the saturation pressure (Psat):

$$C3: P_{H2Oca} \leq P_{sat} \tag{24}$$

Then as depicted in [28], there is a thermodynamic equilibrium between the GdL [29] water content ($\lambda_{GdL}$) and the membrane water content $\lambda_m$ ([30]). The constraint C4 represents this equilibrium. Variables which have an influence on C4 are expressed below ([30]):

$$C4: \lambda_{GdL} = a_1 + a_2 \left( \frac{P_{H2Oca}}{P_{sat}} \right) - a_3 \left( \frac{P_{H2Oca}}{P_{sat}} \right)^2 + a_4 \left( \frac{P_{H2Oca}}{P_{sat}} \right)^3 \tag{25}$$

Regarding the membrane, it has to be hydrated for an appropriate fuel cell operation. An electro osmotic flow (qosm) allows the membrane water supply via a protonic load flow [10]. This flow must get a sufficient value for a good membrane hydration. This is the fifth constraint C5 which involves the electro osmotic flow. Then, a diffusive flow (qdiff) through the membrane also exists and modifies the water content of the membrane. Like the electro osmotic flow, it has to get a relevant value for a good membrane hydration. This value constitutes the sixth constraint C6. The constraint C5 and C6 are expressed as below:

$$q_{H2Oca} = q_{osm} + q_{diff} \tag{26}$$

with:

$$C5: q_{osm} = \lambda_m \tau_0 \frac{I_{fc}}{F} \tag{27}$$

and:

$$C6: q_{diff} = -D_m \frac{\rho_{dry}}{M_m} \frac{d\lambda_m}{dx} \tag{28}$$

An optimal membrane hydration is mandatory for the nominal operation of the PEMFC. Its water content is related to the water concentration inside the membrane. A constraint C7 is thus considered for the membrane water content. This constraint is influenced by the water concentration variation and is expressed as below ([31]):

$$C7: \lambda_m = \frac{M_m}{\rho_{dry}} c_{H2O} \tag{29}$$

where $c_{H2O}$ is the membrane water concentration which depends on the fuel cell temperature.

The membrane hydration also depends on the thermodynamic equilibrium; a relationship between the GdL water content ($\lambda$GdL) and the membrane water content $\lambda_m$ exists. The constraints C5 and C6 are therefore linked to the fuel cell temperature.

The last constraint C8 is also about the membrane water content. Indeed, the lower the membrane water content is, the higher the ohmic resistance is. This resistance can be expressed as below ([31]):

$$C8: \ R_m = \frac{t_m}{\left(b_1 \exp\left(b_2\left(\frac{1}{303} - \frac{1}{T_{fc}}\right)\right)\right)} \tag{30}$$

where, $t_m$ represents the membrane thickness, $b_1$ and $b_2$ are coefficients that depend on fuel cell being tested and $b_1$ depends on the membrane water content ([31]):

$$b_1 = b_{11}\lambda_m - b_{12} \tag{31}$$

Then, the higher the membrane resistance is, the higher the ohmic losses are. The ohmic losses is written as:

$$C8: \ \Delta V_{ohm} = R_m I_{fc} \tag{32}$$

The incidence matrix can now be set in order to create the PEMFC structural graph.

### 5.2. Incidence Matrix of the Structural Analysis

Based on the extraction of the PEMFC constraints, it is possible to create an incidence matrix A ($a_{i,j}$) that allows to link vertices (variables/constraints) and arcs. It contains n rows and m columns:

- $a_{i,j}$ is +1, if the numbered arc j admits the vertex i as origin,
- $a_{i,j}$ is −1, if the numbered arc j admits the vertex i as the arrival,
- $a_{i,j}$ is 0 in other cases.

The incidence matrix is represented in the Table 1.

**Table 1.** Incidence matrix of the structural analysis.

| | C1 | C2 | C3 | C4 | C5 | C6 | C7 | C8 |
|---|---|---|---|---|---|---|---|---|
| $T_{fc}$ | | | 1 | 1 | 1 | 1 | 1 | 1 |
| $q_{O2in}$ | | 1 | | | | | | |
| $q_{H2O}$ | | 1 | | | | | | |
| $q_{hum\_in}$ | 1 | | | | | | | |
| $P_{tot\_in}$ | 1 | | 1 | 1 | | | | |
| $P_{O2\_out}$ | | 1 | | | | | | |
| $P_{O2\_ca}$ | | 1 | | | | | | |
| $q_{H2O\_ca}$ | 1 | 1 | | | 1 | 1 | | |
| $P_{H2O\_out}$ | | 1 | | | | | | |
| $P_{tot\_out}$ | 1 | | 1 | 1 | | | | |
| $q_{H2O\_out}$ | | 1 | | | | | | |
| $q_{O2\_ca}$ | 1 | 1 | | | | | | |
| $P_{H2O\_ca}$ | | 1 | 1 | 1 | | | | |
| $\dot{P}_{H2O\_ca}$ | | 1 | | | | | | |
| $\dot{P}_{O2\_ca}$ | | 1 | | | | | | |
| $V_{fc}$ | | | | | | | | 1 |
| $I_{fc}$ | | | | | 1 | 1 | | 1 |

**Table 1.** *Cont.*

|  | C1 | C2 | C3 | C4 | C5 | C6 | C7 | C8 |
|---|---|---|---|---|---|---|---|---|
| $q_{osm}$ |  |  |  |  | 1 | 1 |  |  |
| $q_{diff}$ |  |  |  |  | 1 | 1 |  |  |
| $\lambda_m$ |  |  |  |  | 1 | 1 | 1 | 1 |
| $\lambda_{GDL}$ |  |  |  | 1 |  | 1 |  |  |
| $C_{H2O}$ |  |  |  |  |  |  | 1 |  |
| $R_m$ |  |  |  |  |  |  |  | 1 |

The control variables are separated from the others with a double vertical line. The structural graph is designed on the basis of the incidence matrix of the Table 1. It is represented on the Figure 6 below.

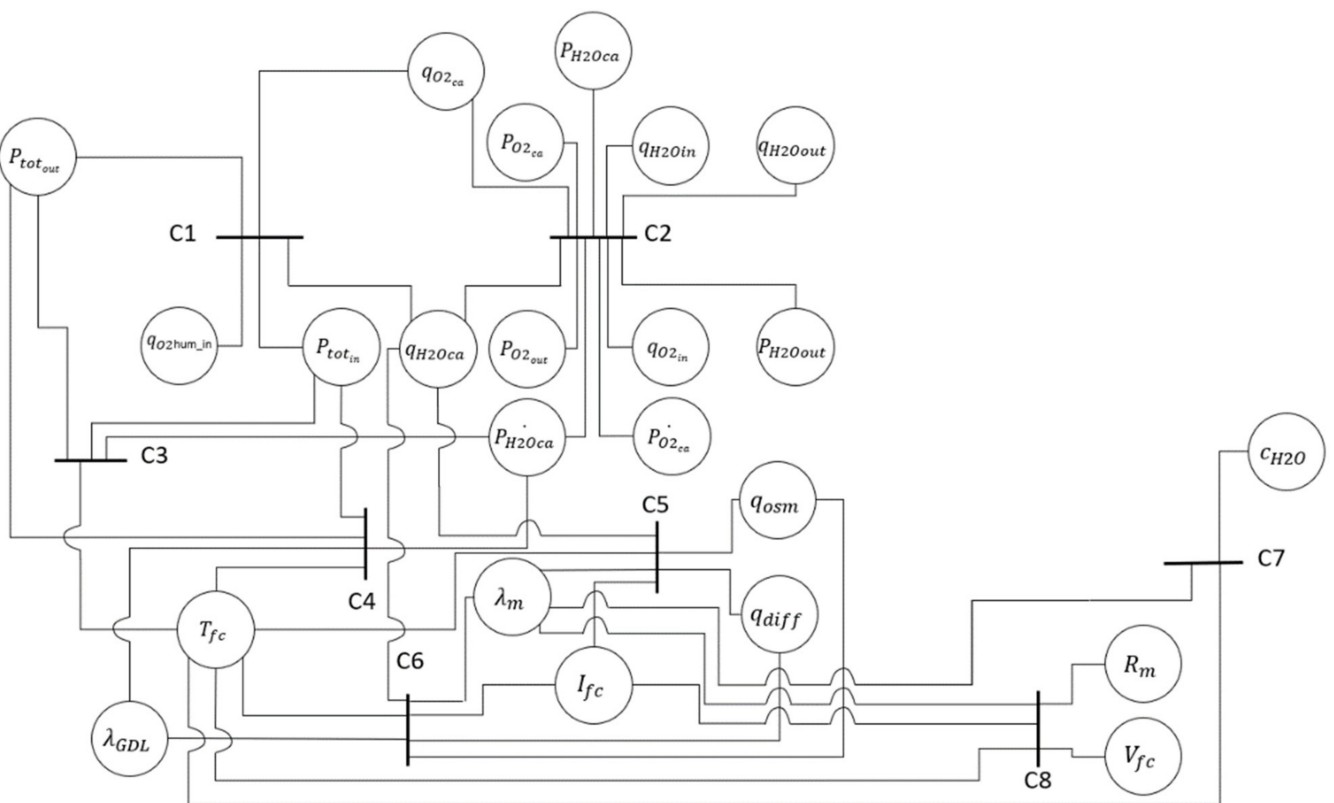

**Figure 6.** Structural graph of the PEMFC.

The next step consists of adding the PEMFC faults on the structural graph in order to represent their interaction inside the PEMFC.

### 5.3. Fuel Cell Flooding Structural Analysis

A PEMFC flooding can occur in two areas. Both inside the GdL with a water droplet accumulation which reduces the catalytic site reactant feeding, and inside the channels by propagation of water droplet accumulation. This water accumulation can also appear directly inside the supply channels and reduce the fuel cell reactant feeding. To integrate the flooding in the structural analysis, the fault is assumed to be a variable which has

influence on the PEMFC constraints defined above. For this purpose, a new variable ($F_{flood}$) which represents a flooding occurrence is added. $F_{flood}$ can therefore be expressed as below:

$$F_{flood} = \frac{V_{electrode\_available}}{V_{geo\_electrode}} \tag{33}$$

with, $V_{electrode\_available}$ is the global volume available in the compartment (channels + GdL) and $V_{geo\_electrode}$ the geometrical volume of the electrode. This variable is set to 1 in case of optimal hydration and to 0 when completely clogged.

The flooding variable has an influence on the supply channels. The new variable is thus added to the constraint C1 which is linked to PEMFC supply channels:

$$C1: q_{O2hum_{in}} = \frac{F_{flood}}{R_{dO2hum_1} + R_{dO2hum_2}} \left( \Delta P + R_{dO2hum_2} q_{O2_{ca}} - R_{dO2hum_2} q_{H2O_{ca}} \right) \tag{34}$$

The flooding variable has also an influence on the GdL and thus on the constraints C2 and C4. These two constraints become:

$$C2: C_h = \frac{F_{flood}}{\dot{P}_{O2_{ca}} + \dot{P}_{H2O_{ca}}} \left( q_{O2_{in}} + q_{H2O_{in}} - q_{O2_{ca}} + q_{H2O_{ca}} - \frac{P_{O2_{ca}} + P_{H2Oca} - P_{O2_{out}} - P_{H2O_{out}}}{R_{dO2hum_2}} + q_{H2O_{out}} \right) \tag{35}$$

$$C4: \lambda_{GdL} = F_{flood} \left( a_1 + a_2 \left( \frac{P_{H2Oca}}{P_{sat}} \right) - a_3 \left( \frac{P_{H2Oca}}{P_{sat}} \right)^2 + a_4 \left( \frac{P_{H2Oca}}{P_{sat}} \right)^3 \right) \tag{36}$$

The incidence matrix is updated with the flooding variable as represented in the Table 2.

**Table 2.** Incidence matrix updated with the flooding variable.

| | C1 | C2 | C3 | C4 | C5 | C6 | C7 | C8 |
|---|---|---|---|---|---|---|---|---|
| $T_{fc}$ | | | 1 | 1 | 1 | 1 | 1 | 1 |
| $q_{O2in}$ | | 1 | | | | | | |
| $q_{H2O}$ | | 1 | | | | | | |
| $q_{hum\_in}$ | 1 | | | | | | | |
| $P_{tot\_in}$ | 1 | | 1 | 1 | | | | |
| $P_{O2\_out}$ | | 1 | | | | | | |
| $P_{O2\_ca}$ | | 1 | | | | | | |
| $q_{H2O\_ca}$ | 1 | 1 | | | 1 | 1 | | |
| $P_{H2O\_out}$ | | 1 | | | | | | |
| $P_{tot\_out}$ | 1 | | 1 | 1 | | | | |
| $q_{H2O\_out}$ | | 1 | | | | | | |
| $q_{O2\_ca}$ | 1 | 1 | | | | | | |
| $P_{H2O\_ca}$ | | 1 | 1 | 1 | | | | |
| $\dot{P}_{H2O\_ca}$ | | 1 | | | | | | |
| $\dot{P}_{O2\_ca}$ | | 1 | | | | | | |
| $V_{fc}$ | | | | | | | | 1 |
| $I_{fc}$ | | | | | 1 | 1 | | 1 |

**Table 2.** *Cont.*

|  | C1 | C2 | C3 | C4 | C5 | C6 | C7 | C8 |
|---|---|---|---|---|---|---|---|---|
| $q_{osm}$ |  |  |  |  | 1 | 1 |  |  |
| $q_{diff}$ |  |  |  |  | 1 | 1 |  |  |
| $\lambda_m$ |  |  |  |  | 1 | 1 | 1 | 1 |
| $\lambda_{GDL}$ |  |  |  | 1 |  | 1 |  |  |
| $C_{H2O}$ |  |  |  |  |  |  | 1 |  |
| $R_m$ |  |  |  |  |  |  |  | 1 |
| $F_{flood}$ | 1 | 1 |  |  |  |  |  |  |

The structural graph with the flooding variable is represented on the Figure 7:

The next step consists in introducing the membrane drying out variable inside the structural analysis. The goal is to identify the constraints which are influenced with its occurrence.

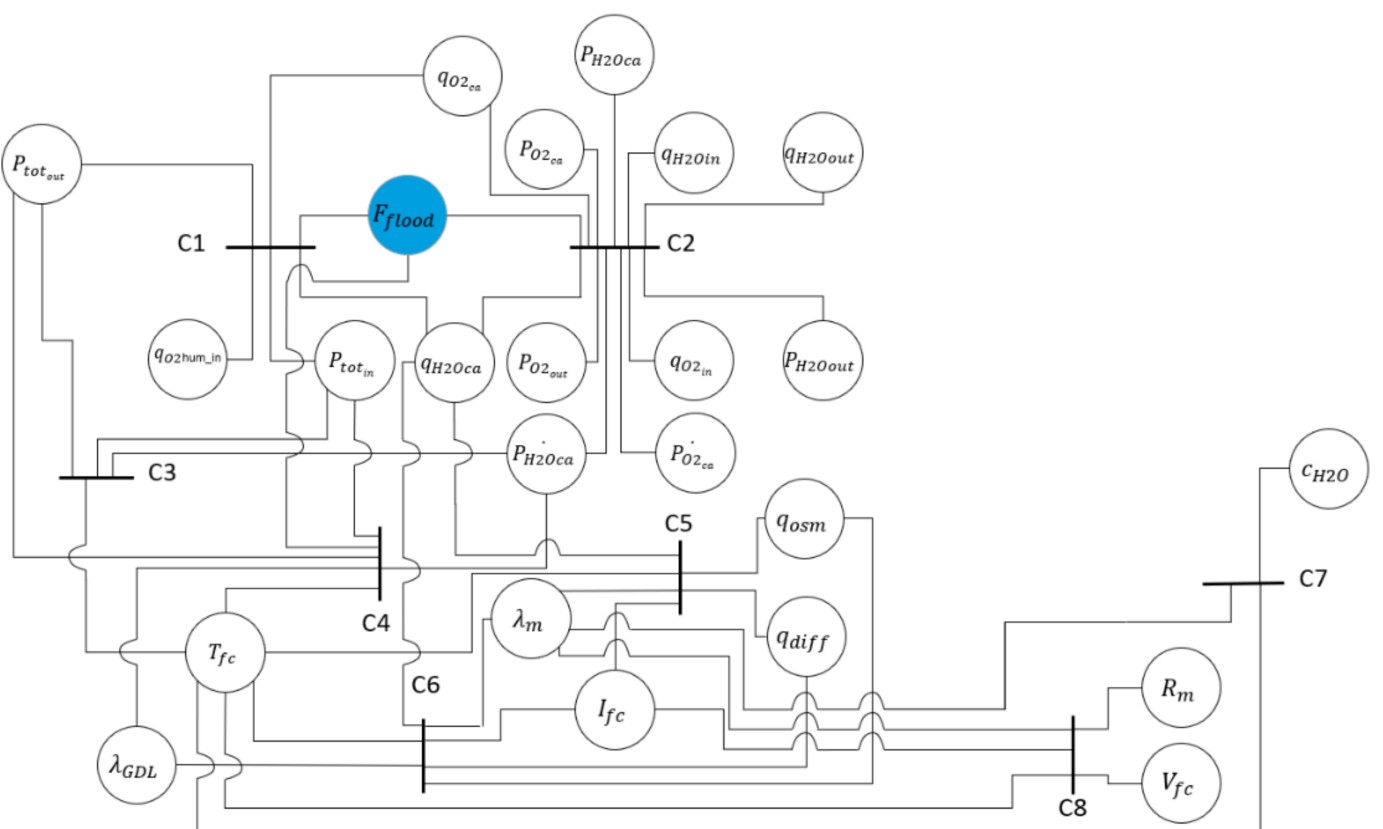

**Figure 7.** Structural graph with the flooding variable.

*5.4. Membrane Drying out Structural Analysis*

The membrane drying out fault results from a decrease of the membrane water content. Therefore, it has an influence on constraints C7 and C8 that involve the membrane water content variable ($\lambda$). The fault can thus be represented by a variable which is expressed as below:

$$F_{dry} = \frac{V_{abs\_membrane}}{V_{abs\_tot\_membrane}} \tag{37}$$

where, $V_{abs\_membrane}$ is the volume absorbed by the membrane and $V_{abs\_tot\_membrane}$ is the total volume absorbable by the membrane. $F_{dry}$ has a value between 0 and 1.

This variable can now be introduced in the constraint C9:

$$C9 : \lambda_m = \frac{M_m \cdot F_{dry}}{\rho_{dry}} c_{H2O} \tag{38}$$

The incidence matrix is updated with the membrane drying out variable as represented on the Table 3.

**Table 3.** Incidence matrix updated with the membrane drying out variable.

| | C1 | C2 | C3 | C4 | C5 | C6 | C7 | C8 |
|---|---|---|---|---|---|---|---|---|
| $T_{fc}$ | | | 1 | 1 | 1 | 1 | 1 | 1 |
| $q_{O2in}$ | | 1 | | | | | | |
| $q_{H2O}$ | | 1 | | | | | | |
| $q_{hum\_in}$ | 1 | | | | | | | |
| $P_{tot\_in}$ | 1 | | 1 | 1 | | | | |
| $P_{O2\_out}$ | | 1 | | | | | | |
| $P_{O2\_ca}$ | | 1 | | | | | | |
| $q_{H2O\_ca}$ | 1 | 1 | | | 1 | 1 | | |
| $P_{H2O\_out}$ | | 1 | | | | | | |
| $P_{tot\_out}$ | 1 | | 1 | 1 | | | | |
| $q_{H2O\_out}$ | | 1 | | | | | | |
| $q_{O2\_ca}$ | 1 | 1 | | | | | | |
| $P_{H2O\_ca}$ | | 1 | 1 | 1 | | | | |
| $\dot{P}_{H2O\_ca}$ | | 1 | | | | | | |
| $\dot{P}_{O2\_ca}$ | | 1 | | | | | | |
| $V_{fc}$ | | | | | | | | 1 |
| $I_{fc}$ | | | | | 1 | 1 | | 1 |
| $q_{osm}$ | | | | | 1 | 1 | | |
| $q_{diff}$ | | | | | 1 | 1 | | |
| $\lambda_m$ | | | | | 1 | 1 | 1 | 1 |
| $\lambda_{GDL}$ | | | | 1 | | 1 | | |
| $C_{H2O}$ | | | | | | | 1 | |
| $R_m$ | | | | | | | | 1 |
| $F_{flood}$ | 1 | 1 | | | | | | |
| $F_{dry}$ | | | | | | | 1 | |

The structural graph with the membrane drying out variable is represented on the Figure 8:

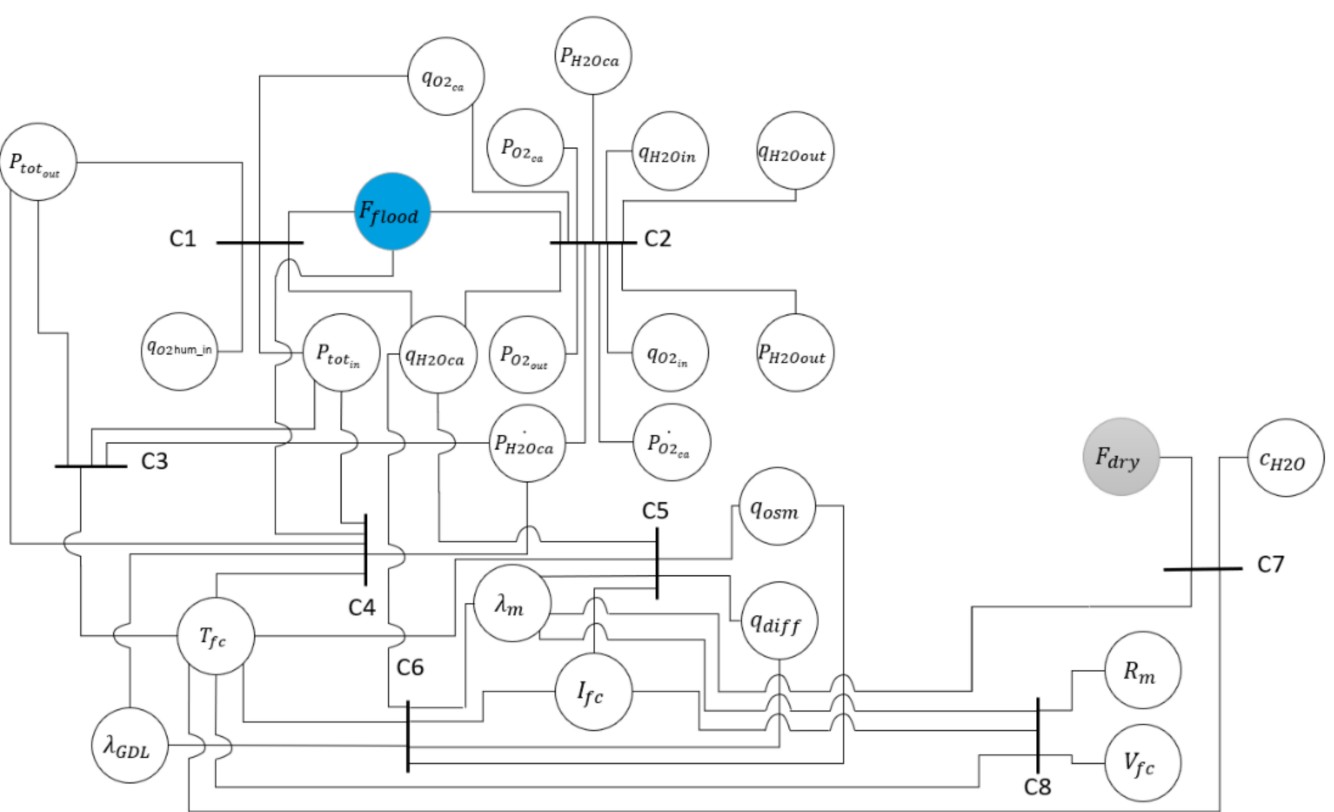

**Figure 8.** Structural graph with the membrane drying out variable.

### 6. FSA in Experimental Context

The experimental FSA illustration is made through the mitigation of fuel cell flooding fault. Indeed, an active fault tolerant control (AFTC) strategy which is detailed in another work [32], is applied on a single cell fuel during a flooding.

AFTC strategies, compared to other kinds of FTC strategies, differs mainly by their structure. Indeed, in a previous literature review [1] it has been highlighted that there are two kinds of FTC strategies: Active or Passive ones. Passive strategies (PFTC) consist of a robust controller design for fault mitigation. In this case, the controller is designed by considering the fault as a disturbance. This structure allows to not use diagnosis tools. But as more is important the number of faults that should be mitigated as more is the PFTC design complexity. To reduce this complexity and instead of use a unique robust controller, the AFTC structure decomposed the mitigation process to several steps. For instance, in [33] authors proposed a three-modules fault mitigation process on a powertrain city bus. Authors used as first module a diagnosis tool for fault detection whereas the second module is composed of decision-making part. Finally, the third module is composed by a set of controllers which implement on the powertrain city bus the mitigation strategy computed by the second module. Another work in [23] where authors proposed another application of the AFTC strategy. In this case the purpose is to address the water management issues. They manage to couple a fault detection and isolation (FDI) algorithm with a reconfiguration mechanism and an adjusting controller. The FDI process is a machine learning based whereas a self-tuning PID is implemented as the control part. Authors also highlight the advantages of the self-tuning PID which shows robustness against noise and model uncertainties. Wu et al. proposed in another fault mitigation process which also consists of an implementation of a three-module AFTC to diagnose a PEMFC fault occurrence, to decide on a relevant mitigation action, and to apply the strategy on the fuel cell through a control set. The main purpose of the method is the distribution of the complexity of the AFTC design into the three modules which significantly simplifies its implementation. In [13] authors proposed a model-based AFTC strategy for PEMFC temperature sensor

fault. They used a FDI process with the aim of real time fault diagnosis with a sliding mode controller for the fuel cell thermal management.

The advantage of the use of AFTC structure lies in its modular aspect. Indeed, it allows the decomposition of the mitigation process into several steps. Each step being dedicating to a different task, it reduces the complexity of the strategy.

### 6.1. Experimental Set Up

The fault tolerant control strategy is applied on a test bench of the FCT brand [34]. It manages to supply a fuel cell with hydrogen at the anode and oxygen or air at the cathode. The experimentations are carried out only with oxygen. The tested fuel cell is a 50 W unit which is composed of N117 ION POWER single-cell of 50 cm$^2$ [35]. The synopsis of test bench is given on the Figure 4.

The Figure 4 shows that the test bench is mounted with two flow rate controllers in order to regulate the input reactant flows. Then, two humification systems are placed on O$_2$ and H$_2$ feeding lines in order to control their relative humidity thanks to two gas temperature controllers. At the event of the fuel cell, two back pressures controllers are used at the anode and cathode lines to manage the fuel pressure. An electronic load is finally connected to the FC electric terminals which can absorb a power of 1.8 kW. The whole test bench is monitored with a LabVIEW virtual instrument.

### 6.2. Flooding Generation Test

The experiment consists of mitigating a flooding occurrence with an AFTC strategy. It will modify the fuel cell operating conditions by changing some control variables iteratively. The selection of the control variables is based on the previous structural graph and on the available sensors and actuators on the test bench. The selected control variables are the: input cathode gas flow rate (q$_{O2ca}$); fuel cell temperature (T$_{fc}$); inlet gas flow relative humidity (q$_{H2Oca}$).

The structural graph shows that the flooding affects the fuel gas channels and the GDL on the fuel gas channels and on the GDL. The control variables q$_{O2ca}$ and q$_{H2Oca}$ have an influence on the gas channels whereas Tfc influences the GDL.

In the experiment, the flooding is generated by introducing liquid water in the cell from the canalization which is between the humidifier and the fuel cell. Indeed, on Figure 4 the gas at the outlet of the humidification system is temperature-regulated in order to reach the relative humidity setpoint. With the temperature controller, it is possible to condense the steam. Then the condensed water goes into the fuel cell to cause a flooding inside the electrode.

The used algorithm for the AFTC strategy is based on an iterative modification of the operating condition with the modification of the selected control variables. The Figure 9 is an illustration of the AFTC mechanism.

### 6.3. AFTC Application to Flooding

The used AFTC strategy has been selected from a previous literature review [1] which is constituted of three modules: diagnosis; decision; control. The diagnosis module is used for the flooding identification based on some outputs fuel cell measurements. Then, if a fault occurred, a decision process is launched through the decision module. At this step, a mitigation strategy is computed for a fast and sustainable fault mitigation. The decision strategy is finally implemented on the fuel cell through the control module which is composed of a set of controllers.

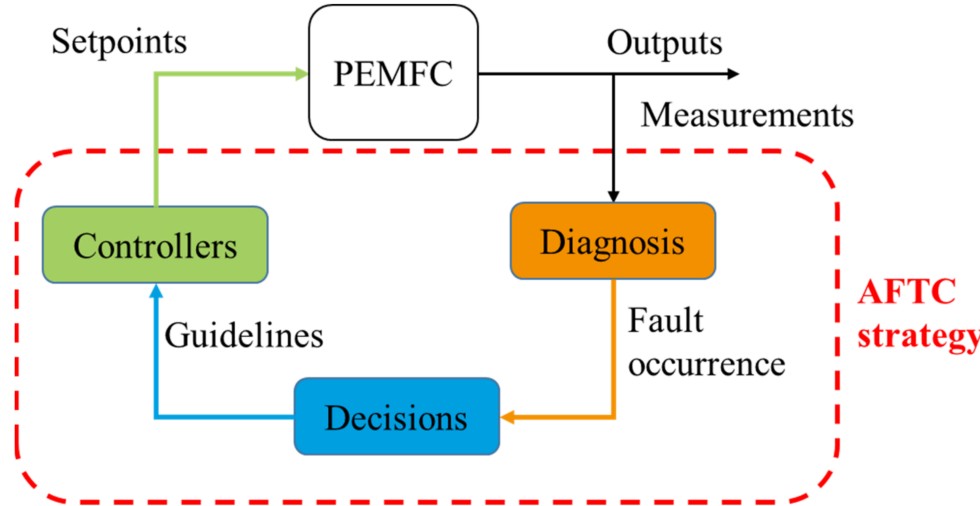

**Figure 9.** Iterative AFTC strategy.

In this work, the diagnosis consists of identifying a flooding occurrence by monitoring the fuel cell pressure drop and the voltage. The decision process provides a decision regarding the fault occurrence to proceed fast mitigation with minimal change in the operating point. The guidelines which are the output of the decision process are transmitted to all controllers to apply the mitigation strategy on the fuel cell.

The AFTC iterative process leads to the setup of two testing cases for the flooding mitigation. Figure 10 represents these testing cases.

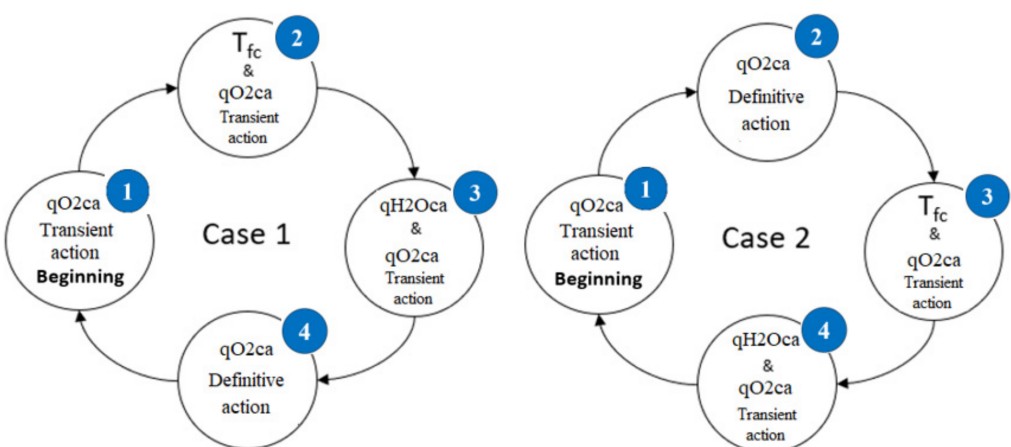

**Figure 10.** Two possible test cases for flooding mitigation.

In both cases, the three selected control variables are used by the AFTC strategy for fault mitigation. If a fault is identified by the diagnosis block, one of the variable values is modified. For instance, in the case 1, the strategy starts with a correction of the gas flow rate qO2ca (1). This is referred to as a Transient action because qO2ca is temporally modified until the flooding is mitigated. The second step of the decision process (2) is an action on Tfc and on qO2ca. Here, Tfc is permanently modified to a new value. In parallel with the action on Tfc, another Transient action on qO2ca is triggered. The third step (3) of the case 1 consists of a permanent modification of the input gas water content (qH2Oca). This action is also supported by a Transient action on qO2ca. The fourth step (4) consists of a permanent change of qO2ca. Here, there is no Transient action.

The test case 2 consists of a different order of the corrective actions. It is composed of the same actions as for the case 1, in a different sequence; except for the first action which is a Transient action on qO2ca.

### 6.4. Experimental Validation of the AFTC Strategy Based on the Variables Extracted from the Structural Graph

A flooding mitigation process based on the test case 1 has been tested on a single fuel cell. Results are depicted in Figure 11a–c.

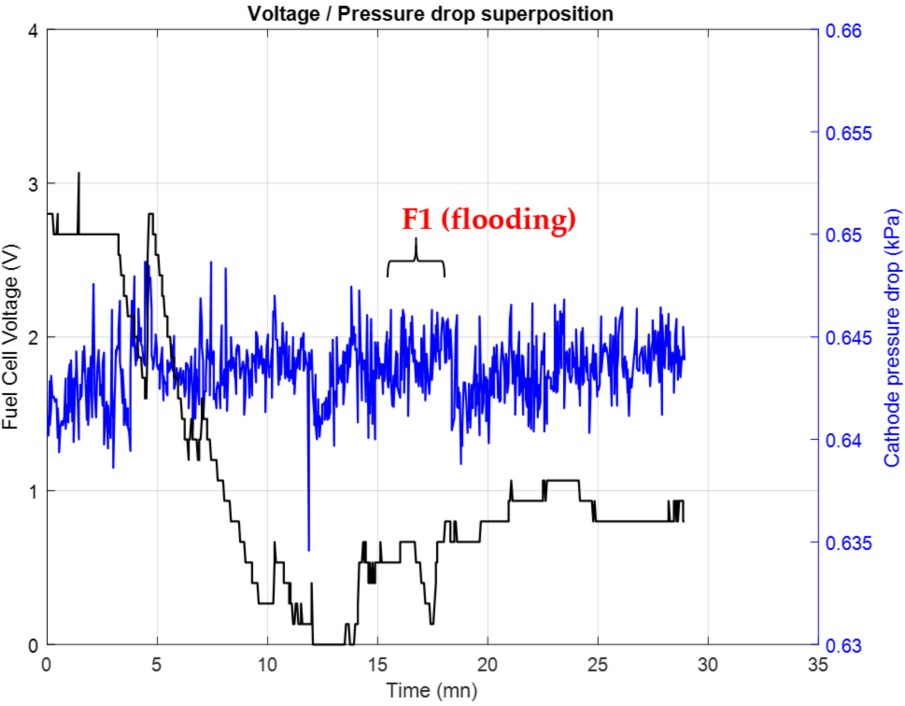

(**a**) Fuel cell voltage and cathode pressure drop superposition

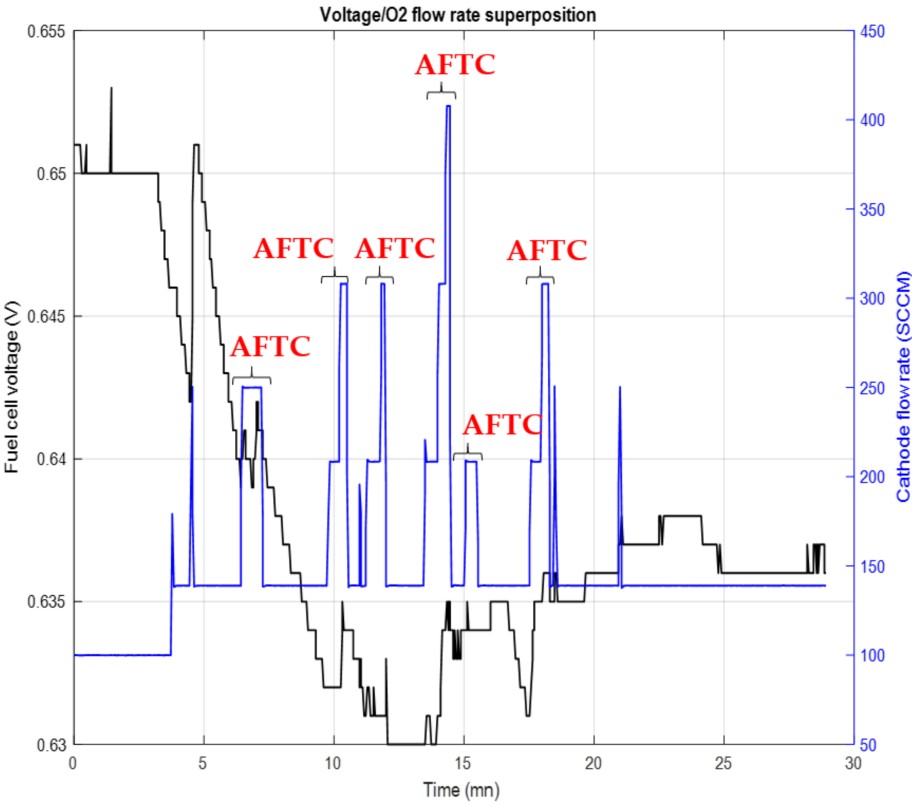

(**b**) Fuel cell voltage and input gas flow superposition

**Figure 11.** *Cont.*

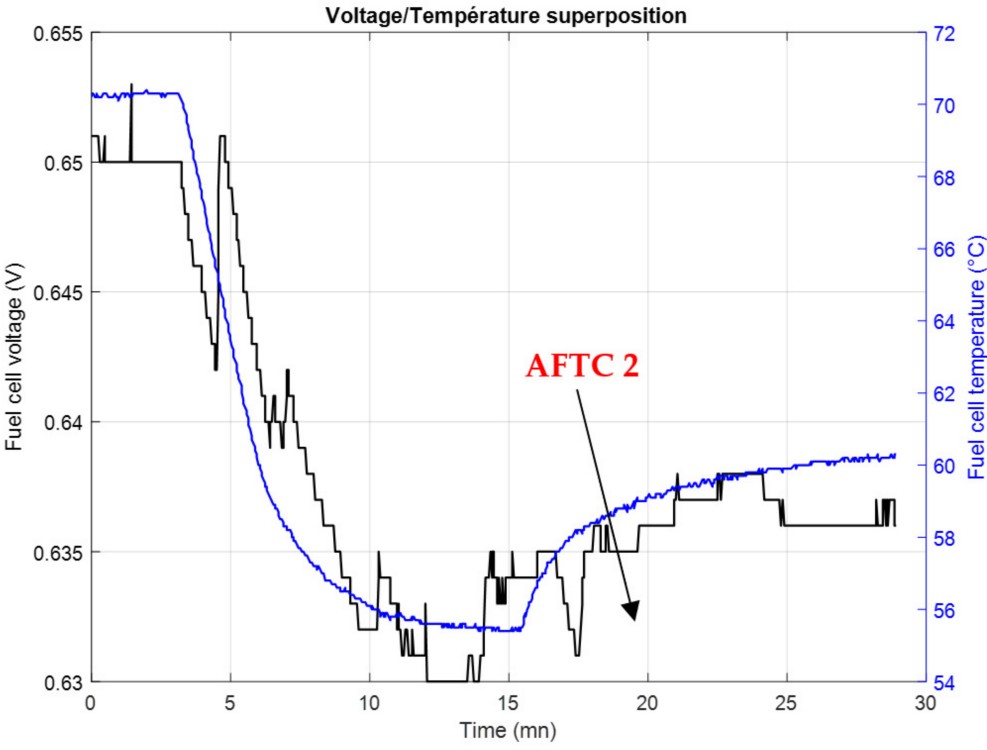

(**c**) Fuel cell voltage and temperature superposition

**Figure 11.** Pressure drop (**a**), oxygen flow rate (**b**) and temperature (**c**) superposition with the fuel cell voltage for the test case 1.

The Figure 11a shows the superposition of the fuel cell voltage with the cathode pressure drop (PD). The PD evolution is used for flooding diagnosis. For instance, F1 corresponds to an increase in PD, which means flooding is in progress.

On Figure 11b all the applied Transient actions on O2 flow rate qO2ca are marked with the AFTC. Each increase in qO2ca aims to mitigate the flooding. When the flooding is mitigated (not diagnosed anymore), the qO2ca value is reset to its value before the Transient action triggering.

On Figure 11c, the second mitigation action of the test case 1, that is the fuel cell temperature is triggered (AFTC 2) at the same time as a Transient action

Figure 12 represents a flooding mitigation process based on the test case 2.

The Figure 12 represents the experimentation applied to the test case 2. Here the first mitigation action is always based on a Transient action, on qO2ca. The second action is also based on the same control variable qO2ca. The increase in qO2ca (AFTC 3) with the Transient action manages to mitigate the flooding by causing the liquid water to drain from the fuel cell.

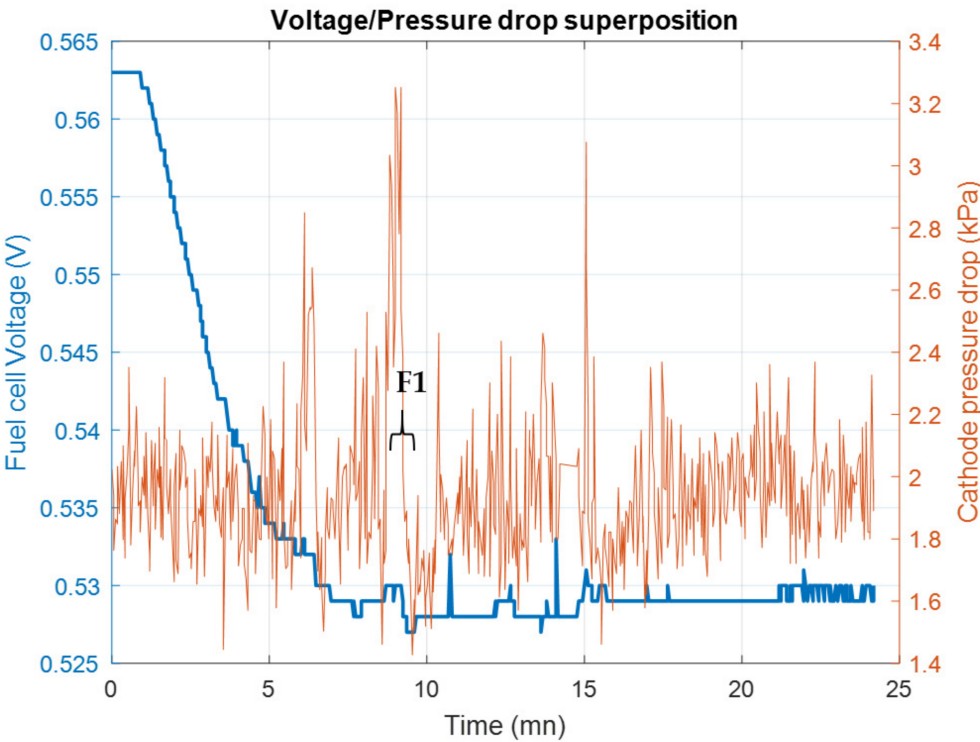

(**a**) Fuel cell voltage and cathode pressure drop superposition

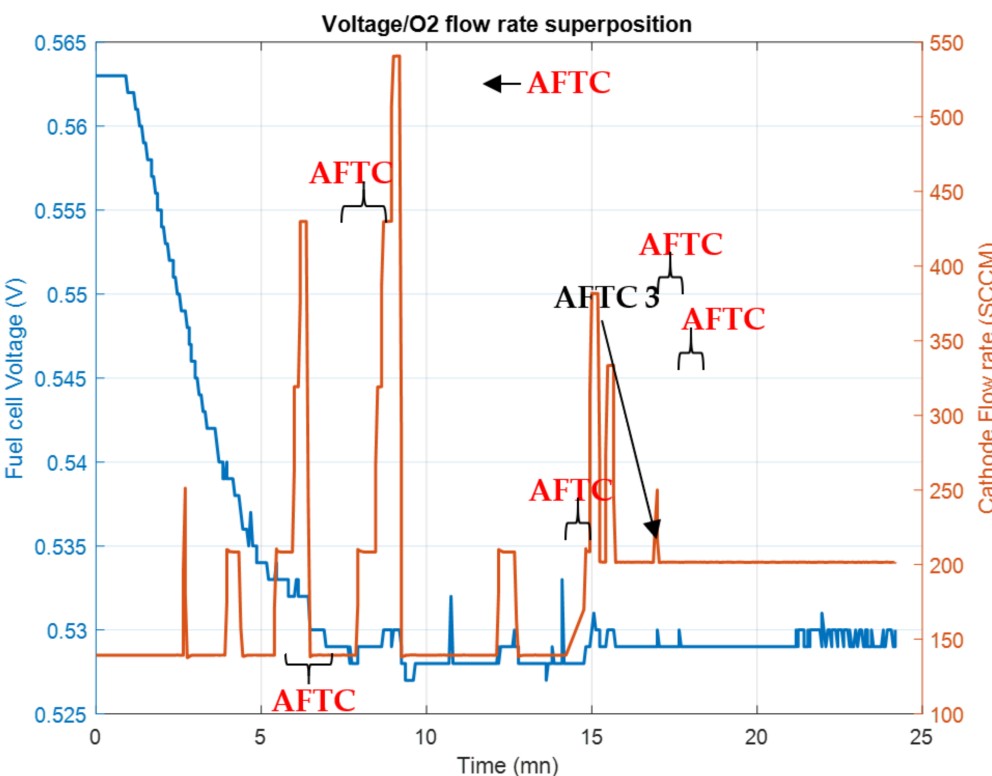

(**b**) Fuel cell voltage and input gas flow superposition

**Figure 12.** *Cont.*

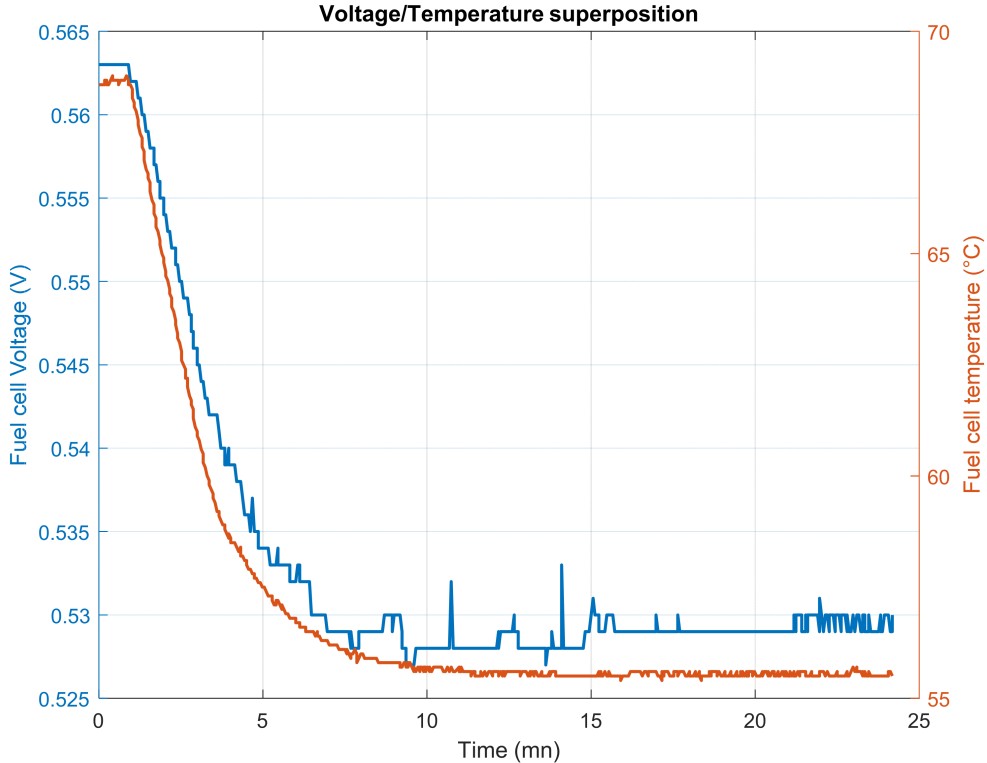

(**c**) Fuel cell voltage and temperature superposition

**Figure 12.** Pressure drop (**a**), oxygen flow rate (**b**) and temperature (**c**) superposition with the fuel cell voltage for the test case 2.

The test results are summarized in the Table 4.

**Table 4.** Test results for flooding mitigation for the tests cases 1 and 2.

| Flooding | Case 1 | Case 2 |
|---|---|---|
| Number of Transient decisions | 6 | 6 |
| Number of permanent decisions | 1 | 1 |
| Permanent decision triggering time | 11 mn | 15 mn |
| Flooding mitigation time | 18 mn | 15 mn |
| Voltage loss | −3.1% | −6.21% |
| Voltage recovery | +1% | +0.35% |

*6.5. Experimental Analysis*

The test of the case 1 and 2 allows the flooding mitigation in about 15 to 18 min and for the same number of Transient actions and permanent actions. The main difference concerns the voltage recovery which is three times fastest in the case of an action on temperature rather than on the cathode flow rate.

Indeed, regarding the structural graph it appears that a flooding acts on the gas feeding channels and on the GDL. Actions on qO2ca allow the water to drain from the fuel cell channels but do not allow the water to drain from the GDL. In the same way, Tfc manages to remove water from the GDL and has no effects on the fuel cell channels. For these reasons, the case 1 which is a mix of actions on the channels and GDL through the qO2ca and Tfc, is a more efficient way for flooding mitigation than case 2 which only acts on the channels.

## 7. Discussion

The structural analysis gives a graphical representation of all PEMFC variables that have an influence on the fuel cell operating conditions. It highlights the coupling of variables inside the PEMFC and leads to the design of a structural graph. The structural graph of the PEMFC on the Figure 8 shows that the fuel cell temperature appears as the most coupled variable. It has an influence on the steam partial pressure at the catalyst site (C3), on the thermodynamic equilibrium of the GdL and the membrane water content (C4), on the electro osmotic and diffusive flows (C5 and C6) and on the membrane water content (C7 and C8). Other variables such as the membrane water flow rate and the input steam flow rate appear to be also strongly coupled. This information, brought by the structural graph, is very relevant to understand the PEMFC complexity and why it is so challenging to maintain fuel cell systems under nominal operating conditions.

Then, the flood variable ($F_{flood}$), which is considered as a fuel cell variable, is integrated in the structural graph. This variable has an influence on the input gas circulation inside the supply channels (C1) and on the GdL gas volume (C2). In case of a design of a flooding mitigation strategy, the structural graph gives the influenced constraints in order to get the appropriate control variables. The structural graph shows that the input gas pressure (Ptot) and flow rate have an influence on the gas feed channel and can lead to the mitigation of the flooding. Regarding a flooding diagnosis tool inside the gas feed channels, the measurement of the inlet gas pressure at the inlet and at the catalytic sites are relevant variables for the design. Indeed, when there is water accumulation inside the gas feed channels, the gas pressure difference between the inlet and catalytic sites increases. However, the pressure at the catalytic site is not measurable and for this reason the pressure difference is determined with the inlet and outlet pressures.

Regarding the flooding in the gas diffusion layer, the analysis conducted gives no relevant variable to diagnose the GdL flooding. This is due to the scale of the model which does not provides the fuel cell behavior inside the GdL and does not discriminate a clogging in the channels or in the GdL. However, the humidified inlet gas and the fuel cell temperature are relevant control variables for the GdL flooding mitigation with a fault mitigation strategy. Indeed, by decreasing the steam injected inside the fuel cell the water accumulation inside the GdL is restrained. An increase of the fuel cell temperature modifies the relative humidity inside the fuel cell and makes the water droplets evaporate.

Concerning the membrane drying out variable on the Figure 8, the structural graph shows that it is related to the membrane water content. The fuel cell current and temperature appear as relevant control variables on the structural graph for a mitigation strategy. These variables have an influence on the membrane water content and can be used in a fault mitigation strategy to alleviate the drying out occurrence. Indeed, the decrease of the fuel cell temperature leads to rehydrate the membrane by an increase of the relative humidity inside the fuel cell. The change of the current value has also an influence on the electroosmotic flow and thus on the water dragged through the membrane which participates to its rehydration. For this faulty condition, the fuel cell voltage, water content and current are relevant variables to set a membrane drying out diagnosis tool.

Therefore, the PEMFC FSA leads to the highlighting of the internal variable coupling. The integration of the fault variables in the structural graph allows understanding the fault process and their location. The analysis underlines the PEMFC constraints that are directly influenced by their occurrence and shows the relevant variables that can be used in FTC strategies.

## 8. Conclusions

The structural analysis approach aims to synthesize the known information about the PEMFC structure and water management issues, leading to flooding and drying out faults. Therefore, the analysis allows describing the system functionalities and their variables. It allows the graphical representation of the fuel cell strong coupling and provides information about relevant variables for the design of a diagnosis tool for flooding and membrane drying out. This analysis also highlights the relevant variables for the design of FTC strategies. Indeed, in case of the development of PEMFC monitoring or mitigation tools, the knowledge of the relevant variables that have to be used and given by the FSA is very helpful. The relevance of the FSA and particularly the graph is shown by the experimental mitigation of a flooding occurrence on a single fuel cell. Indeed, the experiment shows that it is very important to consider which control variable has an influence on which fuel cell functionalities in order to introduce in the FTC strategy enough control variables for fault mitigation.

As discussed in the previous section, the information contained in the FSA depends on the accuracy of the chosen model. But if a higher level was used for the structural graph design, maybe other control variables could appear in the structural graph. For this work and for the used AFTC strategy, only a low level of knowledge was needed because for the FTC design, the fact that not all variables can be measured nor estimated has to be kept in mind to lead to an implementable solution on the used test bench. Hence, there is a trade-off to be found between the different levels of knowledge regarding the selected models.

The literature also reports some other fuel cell faults like CO poisoning, hydrogen and oxygen starvation and low cathode and anode stoichiometry. The next step of the FSA design is therefore to complete the structural graph by adding these faulty conditions.

**Author Contributions:** Conceptualization, E.D., N.Y.S., M.B. and M.-C.P.; methodology, E.D., N.Y.S., M.B. and M.-C.P.; validation, E.D., N.Y.S., M.B. and M.-C.P., B.G.-P.; formal analysis, E.D., N.Y.S. and M.-C.P.; investigation, E.D., N.Y.S., M.B. and M.-C.P.; writing—original draft preparation, E.D.; writing—review and editing, E.D., N.Y.S., M.B. and M.-C.P.; supervision, E.D., N.Y.S., M.B., B.G.-P. and M.-C.P. All authors have read and agreed to the published version of the manuscript.

**Funding:** This research received no external funding.

**Institutional Review Board Statement:** Not applicable.

**Informed Consent Statement:** Not applicable.

**Data Availability Statement:** Not applicable.

**Acknowledgments:** This work has been supported by the Région Réunion and the Région Bourgogne Franche-Comté and by the EIPHI Graduate School (contract ANR-17-EURE-0002).

**Conflicts of Interest:** The authors declare no conflict of interest.

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
