# Peer review of "Fault Structural Analysis Applied to Proton Exchange Membrane Fuel Cell Water Management Issues"

_2673-3293, doi:10.3390/electrochem2040038_

Round 1
Reviewer 1 Report
No
Author Response
Thank you for your reviewing. Other minors modifications have been added in the main text.
Please see the attachment.

Reviewer 2 Report
Objective: to propose a fuel cell fault structural analysis approach which leads to the proposition of a structural graph.
Application: This graph will then be used to highlight the interactions between the control variables and the functionalities of a fuel cell, and therefore emphasize how changing a parameter to mitigate a fault can influence the fuel cell state and eventually cause another fault.
The main issues have been solved:
1) Better highlight the novelty of this paper in Abstract: done.
2) Please clarify the level of accuracy required for fault detection: explained.
3) Simulating some fault conditions would validate the proposed modelling: included.
4) Other recommendations: most were solved.
An active fault tolerant control strategy (AFTC), which is detailed in another work [34], is applied on a single cell fuel during a flooding.
New recommendations:
1) Better highlight the novelty of this paper at the end of Introduction; mention the novelty and main contributions of this research study in a pointwise manner (3-5 bullets).
2) Mention the differences and advantages of the AFTC strategy compared to the FTC approach or Structural analysis extended with active fault isolation; include some relevant references about FTC applied for Fault Detection and Isolation in Fuel Cell Stack System.
3) Table 5-1 (Incidence matrix of the structural analysis) is still a figure: include a table so that the variables are visible and fit in the cells; for this purpose, you can change the rows with the columns.
4) References: Avoid lumped references; include a short critical comment or appreciation for each reference (up to 2 references); see [10-13].
5) Minor editing errors; see for example:
Line 169: Fault Structural Analysis FSA => FSA; the acronym was defined in line 84: Fault Structural Analysis (FSA)
Line 561: active fault tolerant control strategy (AFTC) => active fault tolerant control (AFTC) strategy
Use left indent =0 so that some equations fit on a single line; see for example (5.5).
Etc.
Revise the paper carefully.
Author Response
Reviewer 2:
Dear reviewer,
Thank you for your valuable comments. We have provided answers your question below and revised the paper according to your recommendations. All revisions have been highlighted in main text of the paper and referred to in these answers below.
Comment 1:
“Better highlight the novelty of this paper at the end of Introduction; mention the novelty and main contributions of this research study in a pointwise manner (3-5 bullets).”
Authors response:
We add in the main text at the end of the introduction more information about the novelty of our paper. Please find the modifications from the line 115 to 121.
FSA leads thus to the following contributions:
- Describe the fuel cell structure only with a graph,
- Highlight all variables which influence the fuel cell functionalities and therefore its operating conditions,
- Underline links between the fuel cell functionalities and faults,
- Highlight the relevant control variables which can be used for fault mitigation
Comment 2:
“Mention the differences and advantages of the AFTC strategy compared to the FTC approach or Structural analysis extended with active fault isolation; include some relevant references about FTC applied for Fault Detection and Isolation in Fuel Cell Stack System.”
Authors response:
We have clarified the differences between AFTC strategies and other FTC approaches which are classified in passives strategies (PFTC). These modifications appeared in the main text and highlighted in yellow colour from line 591 to 619.
AFTC strategies, compared to other kinds of FTC strategies, differs mainly by their structure. Indeed, in a previous literature review [1] it has been highlighted that there are two kinds of FTC strategies: Active or Passive ones. Passive strategies (PFTC) consist of the design of a robust controller for fault mitigation. In this case, the controller is designed by considering the fault as a disturbance. This structure allows to not use diagnosis tools. But as more is important the number of faults that should be mitigated as more is the PFTC design complexity. To reduce this complexity and instead of use a unique robust control-ler, the AFTC structure decomposed the mitigation process to several steps. For instance, in [36] authors proposed a three-modules fault mitigation process on a powertrain city bus. Authors, used as first module a diagnosis tool for fault detection whereas the second module is composed of decision-making part. Finally, the third module is composed by a set of controllers which implement on the powertrain city bus the mitigation strategy computed by the second module. Another work in [37] where authors proposed another application of the AFTC strategy. In this case the purpose is to address the water management issues. They manage to coupling a fault detection and isolation (FDI) algorithm with a reconfiguration mechanism and an adjusting controller. The FDI process is a machine learning based whereas a self-tuning PID is implemented as the control part. Authors also highlight the advantages of the self-tuning PID which shows robustness against noise and model uncertainties. Wu et al. proposed in [38] another fault mitigation process which also consists of an implementation of a three-module AFTC to diagnose a PEMFC fault occurrence, to decide on a relevant mitigation action, and to apply the strategy on the fuel cell through a control set. The main purpose of the method is the distribution of the complexity of the AFTC design into the three modules which simplifies significantly its implementation. In [39] authors proposed a model-based AFTC strategy for PEMFC temperature sensor fault. They used a FDI process with the aim of real time fault diagnosis with a sliding mode controller for the fuel cell thermal management.
The advantage of the use of AFTC structure lies in its modular aspect. Indeed, it allows the decomposition of the mitigation process into several steps. Each step being dedicating to a different task, it reduces the complexity of the strategy.
Comment 3:
“Table 5-1 (Incidence matrix of the structural analysis) is still a figure: include a table so that the variables are visible and fit in the cells; for this purpose, you can change the rows with the columns.”
Authors response:
Authors are agree with the reviewer proposition. All incidence matrix has been transformed to tables and rows and columns has been switched. Please, find them at the following lines:
Table 5-1: Incidence matrix of the structural analysis (line 487)
Table 5-2: Incidence matrix updated with the flooding variable (line 538)
Table 5-2: Incidence matrix updated with the membrane drying out variable (line 573)
Comment 4:
“References: Avoid lumped references; include a short critical comment or appreciation for each reference (up to 2 references); see [10-13].”
Authors response:
Each lumped reference has been commented. The modification appears in the main text from the 88 to 103.
For instance, as presented in [10] authors present their strategy for fuel cell fault mitigation. Their work consists of gathering information about the PEMFC state of health through the remaining useful lifetime. The approach is based on the analysis of the system nominal and faulty conditions which are provided by a key variable behaviour. This strategy is thus highly dependent on the relevant chose of the key variable that should be subject to a study of their field of action in the fuel cell for more efficiency. In [11], Yang et al. try to improve the PEMFC reliability with the implementation of a robust fault observer for fault diagnosis the the air management system. Once again, the choose of the estimated variable is a key factor for their diagnosis tool. Indeed, the implementation of their strategy depends on sensitivity of the diagnosis variables to the fuel cell functionalities which are subject to faulty conditions. In [1][12], authors proposed a fuel cell health management system. They used the electrochemical impedance spectroscopy (EIS) in a fault tolerant control strategy in order to diagnose the water management faults. The drawback of EIS diagnosis tools lies in their low computational time, their off line operating mode and the cost of the used equipment. To avoid this problem, the identification of each relevant variables that are the most influenced by each faulty condition for the implementation of a diagnosis tool, can be a solution for their strategy improvement. Another study proposed by Rubio et al. [13], consist of the implementation of a fuzzy model to determine the water dehydration in a PEMFC. The real-time aspect of the strategy involves the use of fast response time of the control variable. The current, the flow rate and the voltage are thus used in the strategy for the fuel cell hydration characterization. This study only considers fast response time variables for the diagnosis tool but the studied phenomena have low, medium and high frequency behaviour. In the case of the introduction of variables which are influenced on the overall spectrum, authors would improve their strategy efficiency.
Comment 5:
“Minor editing errors.”
Authors response:
The acronym redundancies have been deleted as suggested by the reviewer. Moreover, equations are now fitting on singles lines. Please find all modifications about minor editing errors highlighted in yellow colour in the main manuscript text.

This manuscript is a resubmission of an earlier submission. The following is a list of the peer review reports and author responses from that submission.
Round 1
Reviewer 1 Report
The studied topics is important for fuel cell vehicle for system control strategy and prediction of degradation. The paper contents are only written to methods and without solid results and discussion. This is not a complete paper formation for publication. The water management issues of fuel cells are local and transient phenomena, which are related to operating modes (or events). Water flooding is occurring at long-time continuing high current mode and membrane drying is operated idle and low current mode. Those facts did not included in your methods to predictions of water flooding or drying membrane.
Finally, The λGDL could converted to liquid water flooding facult.
Therefore, I suggest total revise this paper.
Reviewer 2 Report
This paper presents a fault structural analysis (FSA) that aims to better identify the sequence of events and the key variables that lead to water management issues in a PEMFC. Key results seem to be an incidence matrix and a structure graph that include flooding and membrane drying variables. Despite the merits of the graphical representation showing the connection between the different variables/operating parameters on a PEMFC system, in the reviewer’s opinion, the results extracted in this study from FSA are not different from a typical mathematical model of a PEMFC. It is well known that temperature as multiple effects on PEMFC performance, as well as reactants flow rate and other operating parameters. Moreover, such effects are not quantified and there is no indication on how to act when a fault (flooding or drying) occurs. Therefore, the reviewer cannot see the utility of the FSA in this study.
Based on the above, the reviewer thinks the manuscript is not suitable for publication in its current form.
Below some comments/corrections/suggestions for the authors reference:
Line 27 – “exogenous and endogenous” instead of “exogenous factor and endogenous”
Line 28 - “and the operating conditions need to be adjusted to mitigate the faults. "
instead of “and need the operating conditions to be adjusted to mitigate the faults. "
line 34 – “variables effects” instead of “variables’ effects”. The wrong use of “ ‘ ” is made many time in the text. Please correct.
Line 35 – FTC not defined
Line 86-87- the meaning of this sentence is not clear.
Line 92 – water management faults is important, but it might not be correct to generalize saying that it is THE MOST important fault of a PEMFC.
Line 105 – a decrease in system performance does not necessarily means degradation. Flooding decreases the performance, but it is generally reversible and thus it does not result into permanent loss. Please clarify what is here considered a fault (degradation of performance only? Permanent degradation?
Line 133 – It seems that there is text missing.
Line 140 – flooding and drying are consequences of improper water management, not causes
Line 157 – what the authors mean by “But this coupling does not appear”?. Not clear.
Line 199 – Differentiation and not Differentiation
Reviewer 3 Report
The structural analysis approach aims to synthesize the known information about the PEMFC structure and water management issues, leading to flooding and drying out faults. Therefore, the analysis allows describing the system functionalities and their variables. It allows the graphical representation of the fuel cell strong coupling and provides information about relevant variables for the design of a diagnosis tool for flooding and membrane drying out.
1) Better highlight the novelty of this paper in Abstract and at the end of Introduction.
The authors stated:
For this work, a low level of knowledge is considered. Indeed, accurate fault model is not mandatory.
But in conclusion, the authors stated that:
As discussed in the previous section, the information contained in the FSA depends on the accuracy of the chosen model. For this reason, the more accurate the model is, the higher the amount of information contained in the FSA will be.
2) Please clarify the level of accuracy required for fault detection.
3) Simulating some fault conditions would validate the proposed modelling.
4) Other recommendations:
Enlarge Figure VI-I and VI-V using left indent = 0; Mention the Incidence matrix as a Table; in fact, it is a Table.
Enlarge Figures VI-II, VI-IV and VI-I using left indent = 0;
This paper thus proposes a fuel cell fault structural analysis approach => This paper makes (presents) a fuel cell fault structural analysis